# Anthropogenic aerosol forcing under the Shared Socioeconomic Pathways
Marianne T. Lund[*,1], Gunnar Myhre[1], Bjørn H. Samset[1]
1 CICERO Center for International Climate Research, Oslo, Norway
*Corresponding author: Marianne T. Lund, m.t.lund@cicero.oslo.no
## Abstract
Emissions of anthropogenic aerosols are expected to change drastically over the coming decades,
with potentially significant climate implications. Using the most recent generation of harmonized
emission scenarios, the Shared Socioeconomic Pathways (SSPs) as input to a global chemistry
transport and radiative transfer model, we provide estimates of the projected future global and
regional burdens and radiative forcing of anthropogenic aerosols under three contrasting pathways
for air pollution levels: SSP1-1.9, SSP2-4.5 and SSP3-7.0. We find that the broader range of future
air pollution emission trajectories spanned by the SSPs compared to previous scenarios translates
into total aerosol forcing estimates in 2100 relative to 1750 ranging from -0.04 W m$^{-2}$ in SSP1-1.9
to -0.51 W m$^{-2}$ in SSP3-7.0. Compared to our 1750-2015 estimate of -0.55 W m$^{-2}$, this shows that
depending on the success of air pollution policies and socioeconomic development over the
coming decades, aerosol radiative forcing may weaken by nearly 95% or remain close to the pre-
industrial to present-day level. In all three scenarios there is a positive forcing in 2100 relative to
2015, from 0.51 W m$^{-2}$ in SSP1-1.9 to 0.04 W m$^{-2}$ in SSP3-7.0. Results also demonstrate significant
differences across regions and scenarios, especially in South Asia and Africa. While rapid
weakening of the negative aerosol forcing following effective air quality policies will unmask
more of the greenhouse gas-induced global warming, slow progress on mitigating air pollution
will significantly enhance the atmospheric aerosol levels and risk to human health in these regions.
In either case, the resulting impacts on regional and global climate can be significant.

## 1 Introduction
Understanding the contribution of aerosols and other short-lived climate forcers to the total
anthropogenic radiative forcing (RF) is becoming increasingly important considering the
ambitious goals of the Paris Agreement. Under scenarios compliant with keeping global warming
below 1.5°C, global greenhouse gas emissions must generally be reduced to net zero by the middle
of the century, placing added focus on the evolution and relative importance of emissions of other
climate-relevant substances for the net future climate impact (IPCC, 2018). Additionally, aerosols
play a key role in shaping regional climate and environment, by modulating clouds, circulation
and precipitation and air quality. In South and East Asia, currently the largest emission source
regions, air pollution is one of the major health risks, estimated to have been responsible for 1.2
million deaths in 2017 in India alone (Balakrishnan et al., 2019). In the same region, aerosols may
have masked up to 1 °C of surface warming (Samset, 2018), and the sensitivity of the regional
climate to reductions in aerosol emissions has been found to be high (Samset et al., 2018a).

Several long-term scenarios for air pollutant emissions exist. Among the most recent examples are the Representative Concentration Pathways (RCPs) (Granier et al., 2011). The RCPs formed the basis for the Coupled Model Intercomparison Project Phase 5 (CMIP5) and have been used in a number of studies to estimate the potential impact of future changes in aerosols on air quality and health (e.g., Li et al., 2016; Partanen et al., 2018; Silva et al., 2016), radiative forcing and temperature (e.g., Chalmers et al., 2012; Shindell et al., 2013; Szopa et al., 2013; Westervelt et al., 2015) and precipitation and other climate variables (Nazarenko et al., 2015; Pendergrass et al., 2015; Rotstayn et al., 2014).

The RCPs were developed to span a range of climate forcing levels and were not associated with specific socio-economic narratives. The RCPs generally reflect the assumption that air quality regulations will be successfully implemented globally (Rao et al., 2017). As a result, emissions of aerosols and aerosol precursors are projected to decline rapidly in all scenarios, even under high forcing and greenhouse gas emission levels. However, despite efforts to control pollutant emissions, ambient air quality continues to be a major concern in many parts of the world. Global emissions of black and organic carbon (BC, OC) have increased rapidly over recent decades (Hoesly et al., 2018). Global emissions of sulfur dioxide ($SO_2$) have declined, driven by legislation in Europe and North America, the collapse of the former Soviet Union and, more recently, air quality policies in China (Li et al., 2017; Zheng et al., 2018). However, in other regions of the world, most notably South Asia, $SO_2$ emissions continue to be high and are increasing. The aerosol and precursor emissions in the RCPs are generated following the assumption that economic growth leads to decreased emissions, i.e., following the so-called environmental Kuznets curve. The real-world representativeness of this relationship has, however, been questioned (Amann et al., 2013; Ru et al., 2018). Combined with the slow observed progress on alleviating air pollution, the question of whether previous projections of future emissions are too optimistic in terms of pollution control arises. More recent scenario development has included alternative assumptions to better understand the mechanisms and interlinkages with reference scenarios and climate policy co-benefits (Chuwah et al., 2013; Rao et al., 2013; Rogelj et al., 2014). These provide a wider range of possible developments but are still largely independent of underlying narratives.

To provide a framework for combining future climate scenarios with socioeconomic development, the Shared Socioeconomic Pathways (SSPs) (O'Neill et al., 2014) were produced. The SSPs provide five narratives for plausible future evolution of society and natural systems in the absence of climate change and combine these with seven different climate forcing targets using integrated assessment modeling, building a matrix of emission scenarios with socioeconomic conditions on one axis and climate change on the other. Associated narratives for air pollution emissions have been developed, representing three levels of pollution control (strong, medium and weak) based on characteristics of control targets, rate of implementation of effective policies and technological progress (Rao et al., 2017). These pollution storylines are then matched with SSP baseline marker and climate mitigation narratives. In SSP1 and SSP5, the combination of strong pollution control, high level of development and increasing health and environmental concerns result in reduced air

pollution emission levels in the medium to long term. A similar, but slower development is seen
under medium control and medium challenges to societal development (SSP2), whereas with weak
control and greater inequality in SSP3 and SSP4 progress is slowed and regionally fragmented.
The numerous drivers influencing future development results in a broad range in projected
emissions, between baseline marker scenarios and for a given SSP depending on the climate
mitigation targets, generally with the highest emissions in SSP3, followed by SSP4, and the lowest
in SSP1 or SSP5 (Rao et al., 2017).
Here we use three of the SSP scenarios as input to a global chemical transport model and offline
radiative transfer calculations (Sect. 2) in order to quantify the future evolution of aerosols under
strong, medium and weak air pollution control. We present results for both global and regional
developments in aerosol loadings and radiative forcing (Sec. 3) and discuss implications of the
findings in the context of previous generation emission scenarios and outlooks for more detailed
studies of the wider climate implications of potential air quality policies (Sect. 4). Conclusions are
given in Sect.5.

2 Method
Atmospheric concentrations of aerosols are simulated with the OsloCTM3 (Søvde et al., 2012).
The OsloCTM3 is a global, offline chemistry-transport model driven by meteorological forecast
data from the European Center for Medium Range Weather Forecast (ECMWF) OpenIFS model.
Here the model is run in a 2.25°x2.25° horizontal resolution, with 60 vertical levels (the uppermost
centered at 0.1 hPa). The present-day aerosol distributions simulated by the OsloCTM3 were
recently documented and evaluated by Lund et al. (2018). We refer to the same paper for detailed
descriptions about the aerosol modules and treatment of scavenging and transport in the
OsloCTM3. All simulations are performed with meteorological data for 2010. Lund et al. (2018)
investigated the impact of meteorology on the simulated aerosol abundances using data for two
years with opposite El Niño–Southern Oscillation (ENSO) index. Differences in global burden of
up to 10% for some aerosol species where found, with occasional larger values in localized regions
over the tropical Pacific and Atlantic Oceans.
Simulations with SSP air pollution emissions from fossil fuel, biofuel and biomass combustion are
performed for the years 2015, 2020, 2030, 2050 and 2100, keeping the meteorology fixed. Nine
emissions scenarios have been gridded and harmonized with the Community Emission Data
System (CEDS) historical emissions (Gidden et al., 2019) and are available via Earth System Grid
Federation (ESGF) by the Integrated Assessment Modeling Consortium (IAMC). Here, we use the
IMAGE (van Vuuren et al., 2017) SSP1-1.9, MESSAGE-GLOBIOM (Fricko et al., 2017) SSP2-
4.5 and AIM (Fujimori et al., 2017) SSP3-7.0 scenarios. SSP1-1.9 represents a pathway with
strong air pollution control, low climate forcing level and low mitigation and adaptation challenges,
while weak air pollution control, high climate forcing and high mitigation and adaptation
challenges characterizes SSP3-7.0. SSP2-4.5 is an intermediate pathway. Air pollution emissions
in the remaining scenarios largely fall in the range between SSP1-1.9 and SSP3-7.0. Each
simulation is run for 18 months, discarding the first six as spin-up. Natural emission sources (soil,
ocean, biogenic organic compounds from vegetation) are kept at the present-day level and the data
sets described in Lund et al. (2018). See Sect. 4 Discussion for comments on the potential
implications of this choice.
Using the same model setup as in the present study, Lund et al. (2018) recently calculated the
historical (1750-2014) evolution of aerosols following the Community Emission Data System
(CEDS) inventory (Hoesly et al., 2018). The future projections from the present study are
combined with this historical time series. Furthermore, whereas Lund et al. (2018) only assessed
the direct aerosol RF, we here include an estimate of the radiative forcing due to aerosol-cloud
interactions.
We calculate the instantaneous top-of-the atmosphere radiative forcing due to aerosol-radiation
interactions (RFari) (Myhre et al., 2013b) using offline radiative transfer calculations with a multi-
stream model using the discrete ordinate method (Stamnes et al., 1988). The same model has been
used in earlier studies of RFari (Bian et al., 2017; Myhre et al., 2013a) with some small recent
updates to aerosol optical properties (Lund et al., 2018). The radiative forcing of aerosol-cloud
interactions (RFaci) (earlier denoted the cloud albedo effect or Twomey effect) is calculated using
the same radiative transfer model. To account for the change in cloud droplet concentration
resulting from anthropogenic aerosols, which alter the cloud effective radius and thus the optical
properties of the clouds, the approach from Quaas et al. (2006) is used. This method has also been
applied in earlier studies (Bian et al., 2017).

3 Results
In the following, we first document the future global emissions and abundances of aerosols,
according to our three chosen SSP scenarios. We then show the resulting regional aerosol burden
levels, and finally global and regional radiative forcing.
Figure 1a-d shows annual global emissions (fossil fuel, biofuel and biomass burning) of BC, OC,
$SO_2$ and nitrogen oxides (NOx) from 1950 to 2100 in the CEDS inventory and the SSPs used in
the present analysis. For comparison, we also include the RCP2.6 (van Vuuren et al., 2007),
RCP4.5 (Smith & Wigley, 2006) and RCP8.5 (Riahi et al., 2007) 2015-2100 emissions. Total
emissions excluding biomass burning are shown in Fig. S1. For all four species, the temporal
evolution and difference between scenarios have similar characteristics. In SSP1-1.9, emissions
are projected to decline from 2015. This decline is particularly rapid for BC and $SO_2$, with
emissions falling to around 25% of their 2015 levels already by 2040. For NOx and OC, emissions
are projected to go down by around 80% by the end of the century. Apart from $SO_2$, SSP3-7.0 sees
an increase in emissions by towards the mid-21[st] century (by 10-20% above 2015 levels), followed
by a decline back to, or slightly below the present-day by 2100. Emissions in SSP2-4.5 follow an
intermediate pathway; a decline throughout the century, but less steep and with a higher end-of-

century levels than SSP1-1.9. As a result of the relatively similar underlying assumptions about the level of air pollution mitigation, the RCPs display much smaller spread and emissions fall throughout the century. All three RCPs generally lie between SSP1-1.9 and SSP2-4.5. There is also a decline in biomass burning emissions in SSP1-1.9 and SSP2-4.5, where emissions are around 30%-40% lower in 2100 compared to 2015. Rao et al. (2017) note that changes in biomass burning emissions are not necessarily driven by air pollution policies but can be linked to assumptions about the land-use sector in the respective integrated assessment models.

The rapidly decreasing anthropogenic emissions in SSP1-1.9 result in global total burdens (Fig. 1e-h) of BC, primary organic aerosol (POA) and sulfate that are 30%, 45% and 60%, respectively, of the present-day level by 2100. Under this pathway, biomass burning sources become relatively more important over the century: fossil fuel and biofuel emissions constitute 70% of the total BC burden in 2015, but only 36% by 2100. Similar end-of-century changes are found under SSP2-4.5, but in this case the decline mainly occurs after 2050. In SSP3-7.0, the global aerosol burdens increase toward the mid-century followed by a small or negligible change to 2100 compared to 2015. The global burden of nitrate is twice as high in 2100 compared to 2015 in SSP3-7.0. This is due to the combination of increased global ammonia ($NH_3$) emissions (not shown here, see Gidden et al. (2019)), which are 30% higher by 2100, a small net change in NOx emissions and a decrease in $SO_2$ emissions, resulting in less competition for available ammonia by sulfate aerosols. The potentially more important role of nitrate aerosols under certain emission pathways has been documented in previous studies as well (Bauer et al., 2007; Bellouin et al., 2011). In SSP1-1.9 and SSP2-4.5, there is negligible net change in $NH_3$ emissions over the century, while NOx emissions decline, resulting in a lower burden also of nitrate. Figure 1i-j shows the simulated anthropogenic global-mean aerosol optical depth (AOD) and absorption aerosol optical depth (AAOD) (calculated as the difference between each year and the 1750 value, with meteorology and hence contribution from natural aerosols constant). The anthropogenic AOD falls from 0.026 in 2015 to 0.0005 in 2100 in SSP1-1.9 and 0.006 in SSP2-4.5. These changes correspond to a reduction of the total AOD of 20% (15%) in 2100 in SSP1-1.9 (SSP2-4.5) from the present-day level of 0.13. In SSP3-7.0, the anthropogenic AOD increases by 12% to 2050 before returning approximately to its present-day value. Similar magnitude decreases in anthropogenic AAOD are found. The decline in anthropogenic AOD is stronger than implied by the burden changes. We note that the sulfate and nitrate burdens include also smaller contributions from natural (ocean and vegetation) sources that remain constant to 2100. In SSP1-1.9 we find a small, negative AAOD value in 2100. This results from emissions on BC and OC being lower in 2100 than in 1750. The stronger decline in anthropogenic AAOD relative to AOD in SSP1-1.9 is reflected in the total (anthropogenic and natural aerosols) Single Scattering Albedo (SSA) (Fig. 1k) which increases to above pre-1970s levels by mid-century and is notably higher than in SSP3-7.0 by the end of the century. As the mechanisms that link aerosol emissions to climate impacts are markedly different for scattering and absorbing aerosols (Ocko et al., 2014; Samset et al., 2016; Smith et al., 2018), this reduction highlights a need for regional studies of aerosol impacts that go beyond the total top-of-atmosphere effective radiative forcing.

The global-mean time series hide significant spatiotemporal differences in aerosol trends. Figure 2 shows the time series of the BC and sulfate burdens, the two dominant species, averaged across

9 regions: North America (NAM), Europe (EUR), Russia (RBU), East Asia (EAS), South Asia (SAS), South East Asia (SEA), North Africa and the Middle East (NAF_MDE), South Africa (SAF) and South America (SAM). The well-known geographical shift in historical emission is clearly reflected, where the largest aerosols loadings were located over North America, Europe and Russia in the 1970s and 80s, but later peaking over Asia. In the coming decades, South and East Asia will continue to experience the highest aerosol loadings under SSP2-4.5 and SSP3-7.0. Towards the end of the century North Africa and the Middle East are projected to experience levels similar to those in South and East Asia. Africa south of the Sahara is presently the third largest BC emission source region (Fig. S2). Under SSP3-7.0, anthropogenic (fossil and biofuel) emissions are projected to increase strongly over the century and the region surpasses East Asia as the largest source in 2100, although levels stay below current emission levels in China. Figure 2 shows that a slightly decreasing BC burden is projected over the region in all three SSPs. In this case, the increase in fossil fuel emissions is offset by a decrease in biomass burning emissions, which constitute a significant fraction of the total BC source here. Despite lower emissions in the latter region, BC burdens in SAF and NAF_MDE are of the same order of magnitude. One reason for this is likely differing scavenging pathways, where aerosols are more effectively removed, and the atmospheric residence time is shorter, further south. Moreover, we note that the regionally averaged burden does not directly link to regional emissions, as they are also influenced by long-range transport. Using multi-model data from the Hemispheric Transport of Air Pollution (HTAP2) experiments, studies have demonstrated that while for most receptor regions, within-region emissions dominates, there are the important contribution from long-range transport from e.g., Asia to aerosols over North America, Middle East and Russia (e.g., Liang et al., 2018; Stjern et al., 2016; Tan et al., 2018). Hence, the projected emission changes in this region can have far reaching impacts.

The radiative forcing of anthropogenic aerosols relative to 1750 is shown for the period 1950 to 2100 in Fig. 3, for RFari, RFaci, and the total aerosol RF (RFtotal), separately. Results from the present study are complimented by results based on simulations from Lund et al. (2018) (see Methods) for the historical period. We calculate a net aerosol-induced RF in 2015, relative to 1750, of -0.55 W m$^{-2}$, whereof -0.14 W m$^{-2}$ is due to aerosol-radiation interactions, as also shown in Lund et al. (2018), and -0.42 W m$^{-2}$ due to aerosol-cloud interactions. Due to the rapid emission reductions projected over the next couple of decades, the RF is less than half in magnitude to its present-day value in SSP1-1.9 already by 2030, continuing to weaken at a slower rate after. In 2100 (relative to 1750), the RFtotal is -0.04 W m$^{-2}$ in SSP1-1.9 and -0.20 W m$^{-2}$ in SSP2-4.5. With emissions following SSP3-7.0, the temporal evolution of RF is nearly flat through the 21$^{st}$ century and is -0.51 W m$^{-2}$ in 2100, only 8% lower in magnitude than in 2015. Even with weak air pollution control (SSP3-7.0) end-of-the-century emissions are slightly lower than the present-day level. Hence, looking only at the period 2015-2100, we estimate a positive aerosol forcing in all three scenarios considered. The RFtotal in 2100 relative to 2015 is 0.51 W m$^{-2}$, 0.35 W m$^{-2}$ and 0.04 W m$^{-2}$ in SSP1-1.9, SSP2-4.5 and SSP3-7.0, respectively. The estimates presented here do not account for the rapid adjustments (or semi-direct effects) associated with BC. Using data from several global models, Stjern et al. (2017) found that the rapid adjustments by clouds offset a significant fraction of the aerosols' positive RFari, reducing the net BC climate impact. In contrast, a recent study by Allen et al. (2019) found positive cloud rapid adjustment. The latter finding would imply

a much stronger non-cloud negative rapid adjustment than presented in Smith et al. (2018) and
methodological differences hence clearly need to be better resolved in order to understand the
contrasting results.
Few modeling-based estimates for comparison with our results exist so far. In a recent study,
Fiedler et al. (2019b) used a simple plume parameterization of optical properties and cloud effects
of anthropogenic aerosols and scaled the present-day aerosol optical depth by the SSP emissions
to derive estimates of future forcing. An effective radiative forcing (ERF) (comparable to our
RFtotal) in the mid-2090s relative to 1850 ranging from $-0.15$ W m$^{-2}$ for SSP1-1.9 to $-0.54$ W m$^{-2}$
for SSP3-7.0 was calculated. This is in reasonable agreement with the estimates derived in the
present analysis, although we find a weaker forcing in SSP1-1.9. Using two idealized scenarios to
span a broader range of emissions than represented in the RCPs, Partanen et al. (2018) also
estimated a broad range in aerosol ERF, from $-0.02$ W m$^{-2}$ to $-0.82$ W m$^{-2}$, in 2100 (relative to
1850). The latter is significantly stronger than our SSP3-7.0 estimate. While not directly
comparable due to differing emission inventories and methodologies, these studies reinforce our
finding that weak air pollution control over the 21$^{st}$ century result in sustained strong negative
aerosol forcing.
The spatiotemporal differences in trend documented above translates into effects on global and
regional RF. In Fig. 4 we therefore show the change in RFtotal over four time periods, 1750-2015,
1750-1990, 1990-2015 and 2015-2030 (for each SSP). Figure S3 show the corresponding results
for RFari and RFaci separately. Whereas the impact of anthropogenic aerosols is a negative RFtotal
everywhere except over the high-albedo deserts and snow-covered regions when taken over the
entire historical period 1750-2015, a positive RF is seen over North America, Europe and Russia
after the 1990-2015 period, driven by decreased $SO_2$ emissions (and somewhat offset by a
simultaneous decline in BC emissions). This positive RF is largely driven by aerosol-radiation
interactions. Over South and East Asia, Africa and most of South America the RFtotal remains
negative, although a significant fraction of the total impact since pre-industrial has already been
realized before 1990. This weaker negative forcing is due to a combination of increasing BC
emissions and a leveling off in $SO_2$ emissions in China in the CEDS inventory (Hosely et al. 2018).
Globally, the combined effect is an increase in global-mean RFtotal over the 1990-2015 period of
$+0.09$ W m$^{-2}$. Using the ECLIPSE emission inventory, Myhre et al. (2017) estimated an increase
in the multi-model RF due to combined changes in aerosols and ozone from 1990 to 2015 of $+0.17$
W m$^{-2}$, with about two-thirds of this from aerosols, i.e., similar to our results using the CEDS/SSP
emissions.
Distinct regional differences are seen also during the period 2015-2030 under the different SSPs.
With emissions following SSP1-1.9, we estimate a positive global-mean RFtotal of 0.33 W m$^{-2}$,
more than three times the RFtotal over the 1990-2015 period. In contrast to the 1900-2015 period,
the strongest RF now comes from aerosol-cloud interactions, as emissions over continental
northern hemisphere regions are low to begin with. The RFtotal is especially large over South and
East Asia, and of opposite sign from what the region has experienced during the past decades.
Smaller positive global mean RFtotal of 0.08 W m$^{-2}$ is estimated also under SSP2-4.5 and SSP3-
7.0 during this period. In contrast to SSP1-1.9, the RF remains negative over India under SSP2-
4.5 and SSP3-7.0 where a continued increase in emissions of $SO_2$ is projected over the next decades.
In all SSPs, the RFtotal over China switches from negative in the past decades to positive over the
2015-2030 period. Recent studies suggest that Chinese $SO_2$ emissions have declined even more
than captured by the CEDS until 2014, indicating that this pattern of forcing may already have
been partly realized (Li et al., 2017; Zheng et al., 2018). In contrast, emissions of India are
projected to increase, at least initially. The potential implications of this feature are discussed in a
separate paper (Samset et al., 2019). Weak RF is found over the African continent in the SSP2-4.5
and SSP3-7.0 scenarios. However, as shown in Figure 2, aerosols will continue to affect local
climate and air quality in this region.

## 4 Discussion

Under a given scenario, emissions of all species generally follow the same global trend, although
the rate of change differs between regions. However, over the recent years, emissions of $SO_2$ have
declined, whereas BC emissions have increased (Hoesly et al., 2018). Considering a hypothetical
and simplified case where the mainly industrial, and perhaps easier to mitigate, $SO_2$ emissions
begin to decline rapidly also in other high emitting regions, whereas the mainly residential, more
challenging, BC sources remain largely unchecked, the aerosol forcing may follow a different path
than estimated here. As an illustrative example, we calculate the contribution to RFari in 2020 and
2050 (relative to 1750) from individual components under SSP1-1.9 and SSP3-7.0 (Table S1).
Taking the sum of the sulfate forcing from SSP1-1.9 and the remaining components from SSP3-
7.0, the total RFari is -0.018 W $m^{-2}$ in 2020, i.e., significantly weaker than when all emissions
follow SSP1-1.9, and 0.15 W $m^{-2}$ in 2050. Continuing along the recent emission development of
declining $SO_2$ emissions and increasing BC could imply a different development in the total
aerosol effect relative to pre-industrial than shown by the three scenarios here, at least towards the
mid-century. We emphasize that these numbers are meant to be illustrative and note that significant
uncertainties surround the climate impact of both BC and the co-emitted organic aerosols. As noted
above, our estimates do not account for the rapid adjustments which might reduce the global
surface temperature response to BC perturbations. Additionally, the role of absorption by so-called
brown carbon remains an important uncertainty (Samset et al., 2018b). Previous work has also
pointed to the possibility of substantial increases in radiative forcing by nitrate over the 21st
century (Bauer et al., 2007; Bellouin et al., 2011; Shindell et al., 2013), and a notable increase in
nitrate burden is also estimated in the present study when emissions follow SSP3-7.0. This
translates into a nitrate RF that is almost a factor 2 stronger in 2100 than in 2020 and constitutes a
correspondingly larger fraction of the RFtotal in this scenario (Table S1)
We present projected future aerosol RF based on single-model simulations. Aerosols, however,
remain one of the most uncertain drivers of climate change, with significant model spread resulting
from several factors, including differences in the simulated aerosol distributions, optical properties
and cloud fields. Myhre et al. (2013a) calculated a present-day aerosol RFari (relative to 1850)
varying from -0.016 W $m^{-2}$ to -0.58 W $m^{-2}$ between 16 global models participating in the AeroCom
Phase II experiment. Prescribing the distribution of anthropogenic aerosols, optical properties and
effect on cloud droplet number concentration in six Earth System Models, Fiedler et al. (2019a)

find a model spread in aerosol ERF of -0.4 W m$^{-2}$ to -0.9 W m$^{-2}$. Among the important consequences of high aerosol forcing uncertainty is the challenge it poses for estimating climate sensitivity. While in a scenario with declining aerosol emissions, combined with an increase in greenhouse gases, the uncertainty in the total anthropogenic forcing can be expected to decrease substantially even without scientific progress (Myhre et al., 2015), the high emission SSP3-7.0 pathway suggest that aerosols may continue to be a confounding factor for constraining climate sensitivity.

While the scope of the present analysis is limited to radiative forcing, the calculated spread in end-of-century forcing under the SSPs will translate into a wide range of possible climate impacts. A number of studies have examined the future aerosol-induced radiative forcing and climate impacts using the RCP projections; see e.g., Westervelt et al. (2015) for a summary of papers published until 2013. While the magnitude of both present-day and future estimates differs between studies, the general characteristic is a significant weakening of the aerosol RF until 2100 in all scenarios. Other studies have investigated the potential for this rapid decline to drive near-term warming (Chalmers et al., 2012; Gillett & Von Salzen, 2013). However, while Chalmers et al. (2012) find a higher near-term warming in RCP2.6 than in RCP4.5 despite lower greenhouse gas forcing in the former, suggesting an important impact of falling aerosol emissions, Gillett and Von Salzen (2013) find no evidence that aerosol emissions reductions drive a particularly rapid near-term warming in this scenario. This points to the importance of inter-model differences in the response to aerosol perturbations. Under SSP1-1.9, aerosol emissions are projected to decline even more rapidly than in RCP2.6 over the coming couple of decades (Fig. 1). If in fact associated with a rapid warming, this development could further hinder the realization of the already ambitious temperature goals of the Paris agreement and this feature hence needs to be better quantified. Previous work also demonstrate effects of falling aerosol emissions also other climate variables such as mean and extreme precipitation (Navarro et al., 2017; Pendergrass et al., 2015) and atmospheric dynamics (Rotstayn et al., 2014). The numerous and significant impacts of aerosols underline the need to encompass the full range of projected emissions, regionally and globally, in future assessment, in particular in light of the crucial role of aerosols in shaping regional climate, regional assessments are needed to capture the impact of different trends.

It is well-established that future changes in aerosols will critically affect local air quality. Silva et al. (2016) found avoided premature mortality in 2100 of between -2.39 and -1.31 million deaths per year for the four RCP. Partanen et al. (2018) estimated almost 80% fewer PM2.5-induced deaths per year in 2100 under RCP4.5 compared to 2010. In contrast, an idealized high aerosol scenario resulted in 17% increase in premature mortality by 2030. These numbers where estimated using present-day population density. Under all SSPs, considerable increases in population density is projected in Africa, the Middle East and South Asia (Jones & O'Neill, 2016) – regions that are also identified as hotspots for exposure and vulnerability to multi-sector climate risk (Byers et al., 2018). In the present study, we estimate an increase in the average surface concentration of anthropogenic aerosols (i.e., BC, POA, sulfate and fine mode nitrate) of 17% and 25% by 2100

under SSP3-7.0 in South Asia and North Africa plus the Middle East, respectively. Air pollution issues are not limited to developing countries. While all scenarios project reductions in surface aerosol concentrations in Europe, North America and Russia, there are substantial differences in the magnitude, from 35-20% lower by 2100 in SSP3-7.0 to around 70% lower in SSP1-1.9, highlighting the potential for further air quality improvements globally.

Our estimates of RFaci exclude contributions from cloud lifetime changes. The estimates of cloud lifetime effect are generally lower in recent studies than in early work, but still give non-negligible contribution to the aerosol forcing (Storelvmo, 2017). Our study does not account for potential impacts of climate change on circulation, precipitation or chemistry, which can affect the lifetime and transport pathways, as well as emissions, of the aerosols. For instance, Bellouin et al. (2011) found increasing atmospheric residence times over the 21[st] century as wet deposition rates decreased. Including both changing climate and emissions, Pommier et al. (2018) suggested that concentrations of particulate matter ($PM_{2.5}$) will increase by up to 6.5% over the Indo-Gangetic Plain to 2050, driven by increases in dust, particulate organic matter and secondary inorganic aerosols through changes in precipitation, biogenic emissions and wind speed. Hence, by keeping natural sources of emissions fixed at present-day levels, our results may underestimate the future aerosols loads. Moreover, a recent review of climate feedbacks on aerosol distributions suggests that in regions where anthropogenic aerosol loadings decrease, the impacts of climate on the variability of natural aerosols increase (Tegen & Schepanski, 2018). Changing climatic conditions may also affect the radiative forcing through changing cloud distributions and surface albedo. While our approach clearly disentangles and assesses the changes in aerosols resulting from changes in anthropogenic emissions, representation and knowledge of feedback processes are important for understanding the full role of future aerosols in the climate system.

## 5 Conclusions

Using a global chemistry transport model and radiative transfer modeling, we have estimated the projected future loading and radiative forcing of anthropogenic aerosols under the most recent generation of scenarios, the Shared Socioeconomic Pathways. These new air pollution scenarios link varying degrees of air pollution control to the socioeconomic narratives underlying the SSPs and climate forcing targets, spanning a much broader range of plausible future emission trajectories than previous scenarios. Here we have used three scenarios: SSP3-7.0 (weak air pollution control, high mitigation and adaptation challenges), SSP2-4.5 (medium pollution control, medium mitigation and adaptation challenges) and SSP1-1.9 (strong pollution control, low mitigation and adaptation challenges). In all three scenarios, we estimate a positive aerosol forcing over the period 2015-2100, although with very different timing and magnitude depending on stringency of air pollution control. The end-of-century aerosol forcing relative to 2015 is 0.51 W m$^{-2}$ with emissions following SSP1-1.9, 0.35 W m$^{-2}$ in SSP2-4.5 and 0.04 W m$^{-2}$ in SSP3-7.0. While effective air pollution control and socioeconomic development following SSP1-1.9 results in a rapid weakening of the aerosol RF compared to the pre-industrial to present-day level already by 2030, there is little change in the global mean aerosol forcing over the 21[st] century in a

regionally fragmented world with slower mitigation progress (SSP3-7.0). Significant spatiotemporal differences in trends are also highlighted. Most notably, under weak air pollution control, aerosol loadings in East and South Asia temporarily increase from present levels but start to decline after 2050 and return to current levels of slightly below by 2100. North Africa and the Middle East reaches the levels of South Asia by the end of the century and there is no declining trend this century. The present analysis is limited to the documentation of radiative forcing and aerosol loads. Under both rapidly declining and sustained high emissions, aerosols will play an important role in shaping and affect regional and global climate.

## Code availability

Oslo CTM3 is stored in a SVN repository at the University of Oslo central subversion system and is available upon request. Please contact m.t.lund@cicero.oslo.no. In this paper, we use the official version 1.0, Oslo CTM3 v1.0.

## Data availability

The gridded SSP anthropogenic emission data are published within the ESGF system https://esgf-node.llnl.gov/search/ input4mips/ (last access: December 2018). Model output and post-processing routines are available upon request from Marianne T. Lund (m.t.lund@cicero.oslo.no).

## Author contributions

MTL performed the Oslo CTM3 experiments and led the analysis and writing. GM performed the radiative transfer modeling and BHS contributed with graphics production. All authors contributed during the writing of the paper.

## Acknowledgements

The authors acknowledge funding from the Norwegian Research Council through grants 248834 (QUISARC) and 240372 (AC/BC). We also acknowledge the Research Council of Norway's programme for supercomputing (NOTUR).

## Competing interests

The authors declare that they have no conflict of interest.

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

## Figures


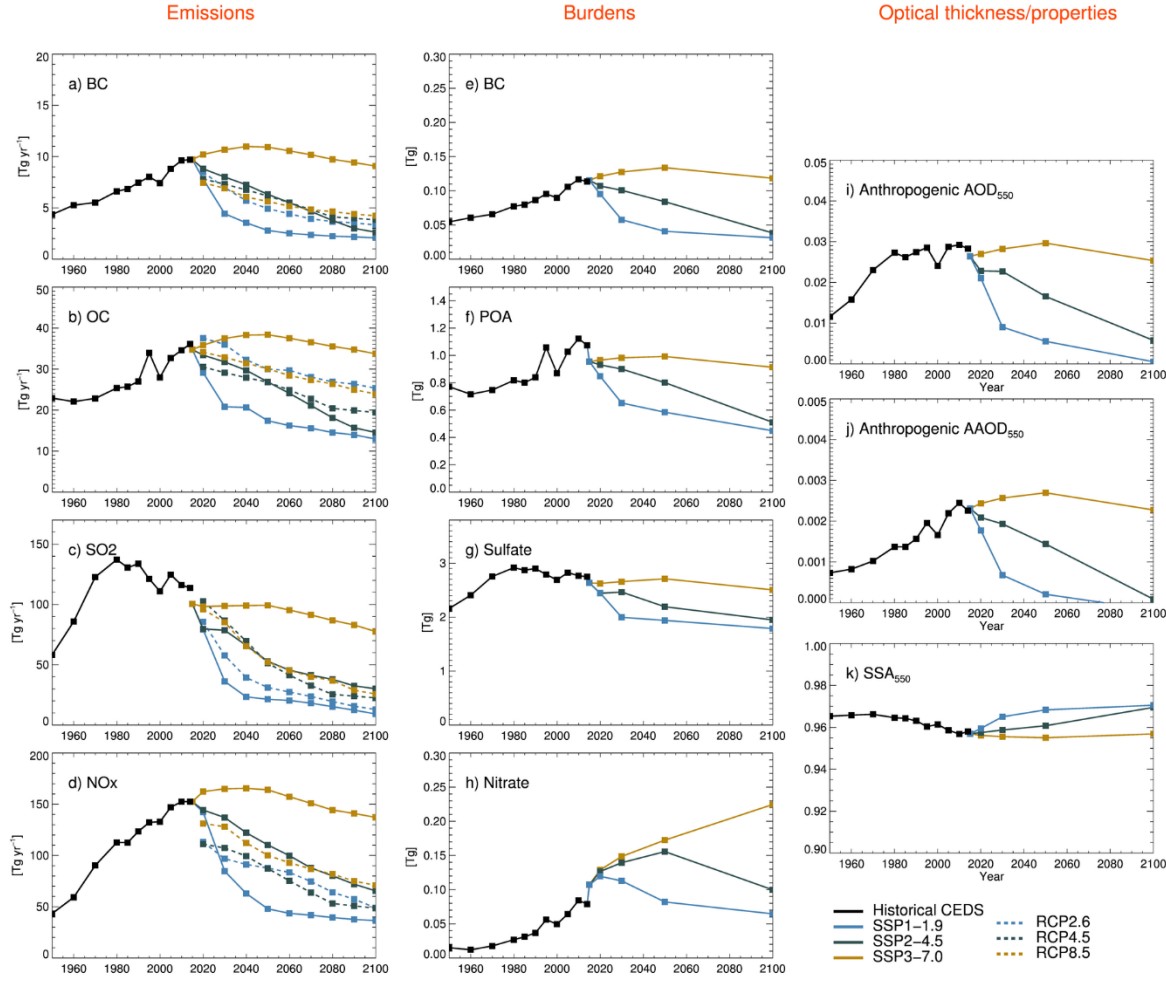


*Figure 1. Left: Annual global emissions (fossil fuel, biofuel and biomass burning) of BC, OC, SO₂ and NOx over the period 1950 to 2100 from the CEDS historical inventory and SSP1-1.9, SSP2-4.5 and SSP3-7.0 (solid colored lines). Emissions from RCP2.6, RCP4.5 and RCP8.5 (dashed lines) are added for comparison. Middle: Modeled total global burdens of BC, POA, sulfate and fine mode nitrate. Right: Anthropogenic AOD and AAOD, and total (anthropogenic and natural) SSA at 550nm.*





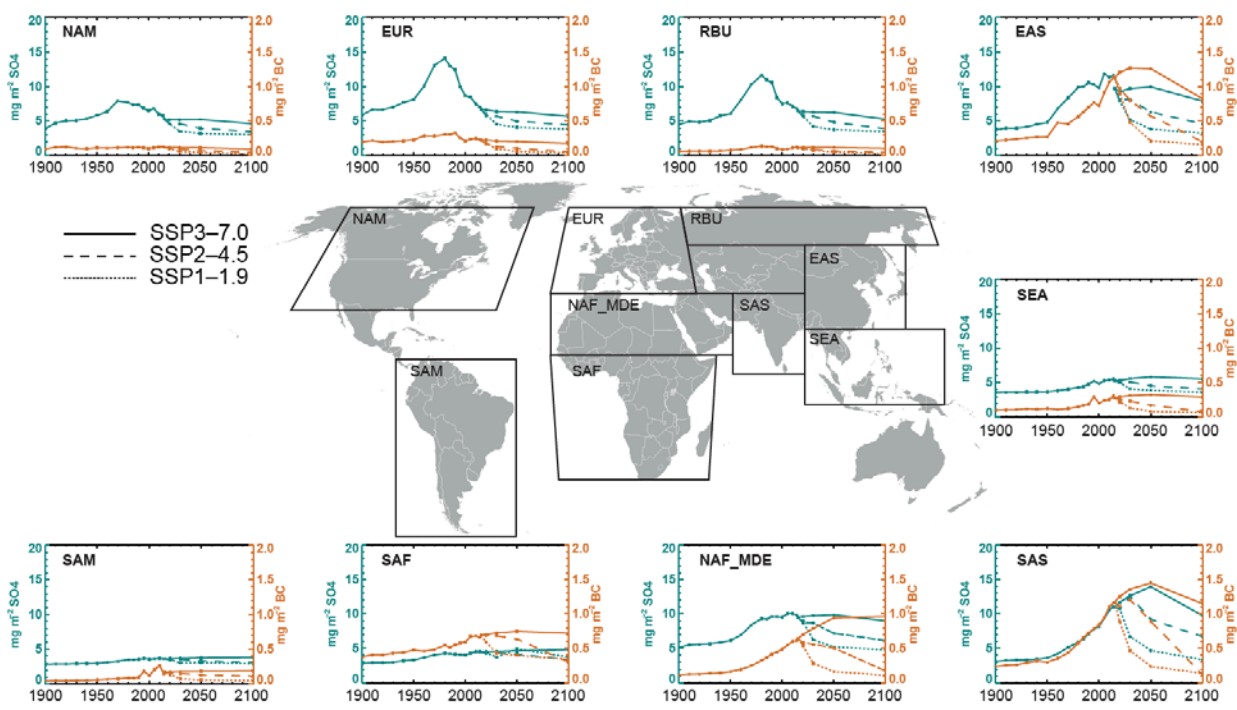


*Figure 2: Regionally averaged burdens of BC and sulfate aerosols from 1900 to 2100 using CEDS historical emissions and SSP1-1.9, SSP2-4.5 and SSP3-7.0.*




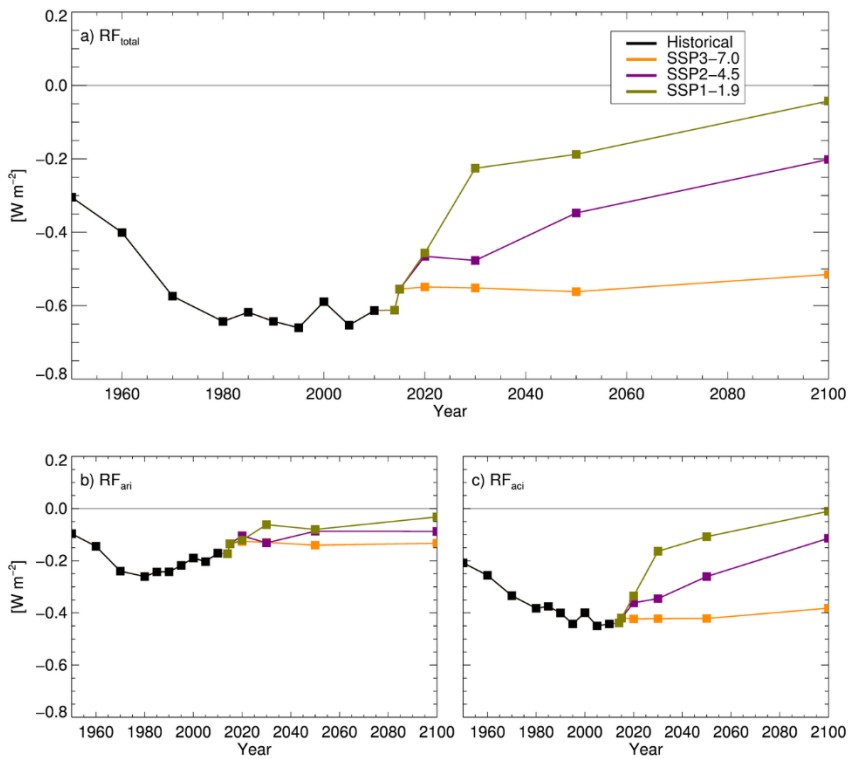


*Figure 3: Radiative forcing of anthropogenic aerosol 1950-2100 relative to 1750: a) total aerosol RF (RFtotal), b) aerosol-radiation interactions (RFari) and c) aerosol-cloud interactions (RFaci).*




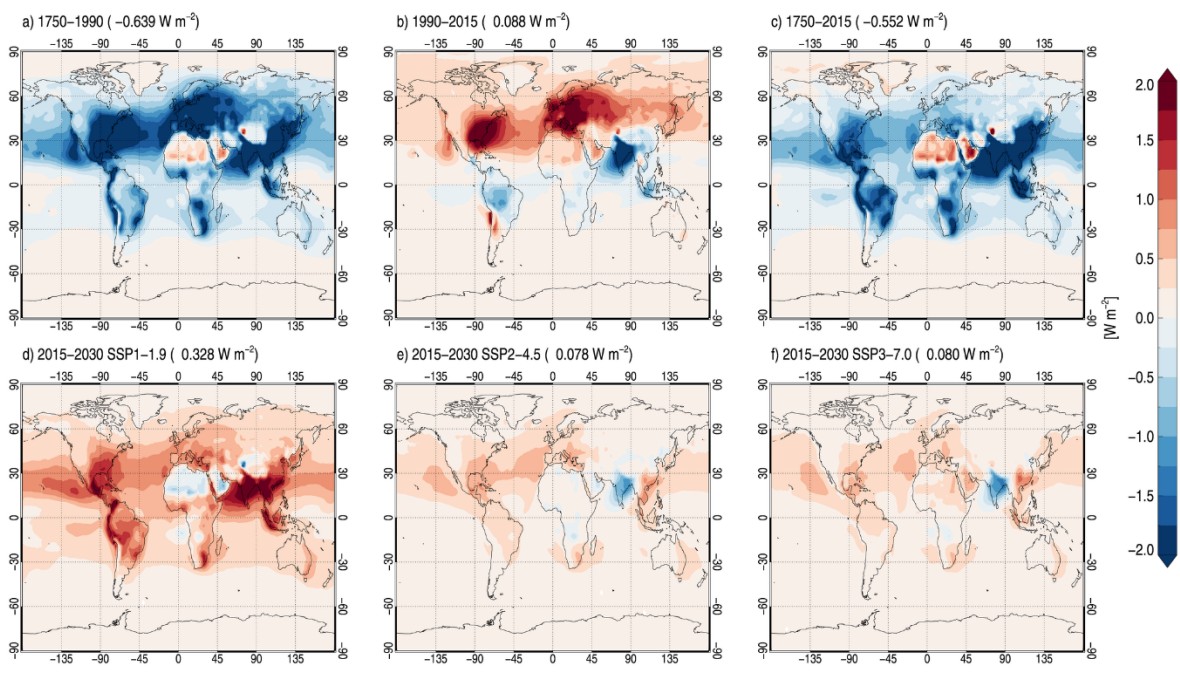


*Figure 4: Total aerosol RF over four time periods: 1750-1990, 1990-2015, 1750-2015, and 2015-2030 for*
*each of the three SSP scenarios considered here.*