# Peer review of "Anthropogenic aerosol forcing under the Shared Socioeconomic Pathways"

_Atmospheric Chemistry and Physics, 2019_

## Referee Comment (RC1) · Anonymous Referee #1 · 24 Jul 2019

In this paper, Lund et al estimate the future loading and radiative forcing of anthropogenic aerosols under three of the SSP emission scenarios. This is a good study in my opinion that merits publication in ACP following minor revisions as described in my detailed line-by-line comments below.

L18-19: The differences between 2100 and 2015 reported here should equal the 1750-2100 values (given in L14-15) minus the 1750-2015 value (-0.61 W/m2) I'd expect, but they don't. Seems like a mistake and accidentally the opposite of the 1750-2100 values are given here instead of the differences for 2100 vs 2015 which I'd think are +0.57 and +0.10 W/m2, respectively (unless the values in L14-15 are wrong). Or if there is some non-linearity or incompatibility between the various estimates please revise so this is not so confusing to the reader.

[Figure]

L43-44: Rather than the Li et al study cited here for air quality/health, which looked only at one country using one model, one of the worldwide multi-model studies by Silva et al would be a better choice (e.g. Silva et al, The effect of future ambient air pollution on human premature mortality to 2100 using output from the ACCMIP model ensemble, Atmos. Chem. Phys., 16, 9847-9862, 2016.)

Similarly, rather than (or in addition if you like) the Szopa et al and Westervelt et al studies, each of which looked at just one model, the reader could more usefully be pointed to the multi-model studies such as Shindell et al., Radiative forcing in the ACCMIP historical and future climate simulations, Atmos. Chem. Phys., 13, 2939–2974, 2013; and Rotstayn et al, Why does aerosol forcing control historical global-mean surface temperature change in CMIP5 models?, J. Climate, 28, 6608-6625, 2015.

L56: Clearer to write as "continue to be high and are increasing" than "also continue to be high and increasing" as the 'also' seems not to relate to the previous sentence (about SO2) but an earlier one so is hard to follow.

L56-58: It would be good to include additional explanation here that the RCP aerosol emissions were based on the assumption that economic growth leads to decreasing emissions of precursors, a so-called environmental Kuznets' curve/behavior. An excellent discussion of the background to assumptions such as those is given in Amann et al, 2013. Regional and global emissions of air pollutants: recent trends and future scenarios. Annual Review of Environment and Resources, 38, pp.31-55; and a recent paper analysed the link between economic growth and aerosol precursor emissions and indeed supports the discussion here (and in Amann et al) that RCP projections are likely too optimistic as those emissions do not necessarily decline with growth in GDP or follow CO2 (see Ru et al, The long-term relationship between emissions and economic growth for SO2, CO2 and BC, Env. Res. Lett., 13, 124021, 2018.) This discussion would help set the context for the new approach in the SSPs where air pollution controls have independent settings, as described in the next paragraph.
L74: Replace 'there is progress is slowed' with just 'progress is slowed'.

L90: Please also give the number of vertical layers and the model top as that'd help the reader get a general sense of the model in addition to the horizontal resolution.

L150: 'Becomes' -> 'become'.

L186: Delete 'the' before 'South Asia'.

L189: Add 'the' before 'Sahara'.

L201: Add 'that' after 'demonstrated'.

L220-221: The authors describe the conclusion drawn in the Stjern et al study about the semi-direct impact of BC here. They should also include the results of Allen et al., Observationally constrained aerosol–cloud semi-direct effects, npjCAS, 2, 16, 2019 as those suggest the semi-direct effect may not be well captured by the models used in Stjern et al and so drawing conclusions about the sign of the adjustment is less certain than the current text implies.

The same goes for text in L281-282 – might reduce temperature response not the more definitive 'have been shown to'.

L225-230: It would be helpful to be clear about the time periods for the forcing values quoted from these other two studies (for Fiedler, forcing when vs when; for Partanen, it is stated that it's for 2100, but relative to when?).

L269-282: The discussion here starts to delve into the role of individual aerosol species, which is good. I like the current discussion regarding SO2 and BC. I'd like to see a bit more on this, however, in particular I would request that the authors extend their Table S1 to include the year 2100 (data they should already have), and add a discussion of the nitrate forcing towards the end of the century given the large increase in burden shown in Figure 1h for SSP3. That could be usefully compared with the results of the Bauer et al, Bellouin et al, and Shindell et al (see reference in comments on L43-

44 for the latter, the other two are already cited) studies that discussed the possibility of substantial increases in negative RF from nitrate over the 21st century.

L318-320: Again could compare with the Silva et al results for future RCPs (see reference in comments on L43-44).

L331: 'Impose' seems an odd word to use here, as usually things people don't want are imposed upon them whereas improved air quality is something people do want. I suggest changing to 'lead to' or something similar.

L332-347: The Methods section describes how only cloud albedo effects are analysed here (as RFaci is calculated offline). This section should point out that the calculations here not only neglect climate feedbacks, but neglect the entire cloud lifetime portion of RFaci, which may be important.

L363: Useful for those not so familiar with the SSPs to add "(SSP3)" after "regionally fragmented world with slower mitigation progress" here.

L365: "Increases" -> "increase" and "starts" -> "start".

---

## Referee Comment (RC2) · Anonymous Referee #2 · 26 Aug 2019

I have read the paper "Anthropogenic aerosol forcing under the Shared Socioeconomic Pathways" by Lund et al. The paper presents radiative forcing estimates from three selected SSP scenarios. These are a useful set of first results, albeit from one model, of scenarios that will likely be widely used in the future. I have a few comments below for improving/clarifying the presentation of the material, as well as a couple of additional points that should be discussed.

Abstract this text "aerosols under three different levels of air pollution control: strong (SSP1), medium (SSP2) and weak (SSP3). " should be revised, given that far more than air pollution controls impact emission levels in these three scenarios. It would be more accurate to describe these as representing three contrasting projections for air pollutant emission levels.

[Figure]

More context should be given in the introduction when the scenarios are introduced. Its too simplistic to simply call the scenarios simply high/low air pollution. Air pollution controls plus the magnitude of the various drivers of emissions (e.g., population levels, economic growth, rural access to modern energy, GHG emissions policy, etc.) all play a role in determining the ultimate emissions level.

For example, referring to Rao et al. Figure 2, for the Ref case scenarios (e.g. no GHG emissions reduction policy) emissions can differ significantly between SSP1 and SSP5, even though both of these scenarios represent storylines with strong air pollution controls. Similarly, emission levels are generally quite a bit higher in SSP3 as compared to SSP4, even though the emission control assumptions are similar. Note also that the SSPs are from different projection models, which means that one also has be cautious in such comparisons.

General The impact of inter-annual variability should be discussed given that meteorology for just one year is used. How much does the selection of that year influence results?

Line 47 - "generally reflect the assumption that stringent air quality regulations will be successfully implemented globally (Rao et al., 2017"

Suggest replacing stringent with a somewhat more neutral word (perhaps "substantial""). The RCP's represented a somewhat middle of the road air pollutant emission control assumptions, but by no means were they at maximally feasible levels (which is what might be read by "stringent").

The more important point to be made here is that there was limited variation in air pollution control assumptions across the RCP scenarios.

Line 95-105 What was assumed for open burning? These are also supplied in the future scenarios, (but are from van Marle et al 2017, not Hoesly et al., 2018). Note that these are not "natural" emissions, as much of these emissions are due to human

activity.

Line 123 While I understand why this "For simplicity we refer to SSP1-1.9 as SSP1, SSP2-4.5 as SSP2 and SSP3-7.0 as SSP3 throughout the text."is done, however, this is inaccurate and may lead to misunderstanding on the part of readers, as there can be systematic differences even between scenarios with the same storyline.

SSP1-1.9, for example, is a very strong GHG mitigation scenario which means that fossil fuel use is drastically reduced (and what fossil fuel that is used tends to have lower air pollutant emissions). So emissions will tend to be on the low side of what is already a low reference scenario. Emissions can be much lower than the reference case SSP1, particularly in earlier years. Similarly for SSP2-4.5, emissions here can be significantly influenced by the fact that the 4.5 scenario contains policies to limit greenhouse gas emissions.

I suggest first, as mentioned above, that a little additional context be given for these scenarios. It would help, on first introducing the scenarios, that the fuller version of the scenario name that also contains the model name is used. That will help enforce to the readers that these are from different projection models. The presence, or not, of a climate policy should be mentioned (e.g. present in SSP1-1.9 and SSP2-4.5 ) should be mentioned in this introduction, since this can have a strong influence on the emissions pathway.

Some of the figures have the fuller scenario names and some do not. Suggest that all figures have the names that contain the forcing target.

I suggest the fuller scenario names (e.g., SSP1-1.9) be returned to in the discussion and conclusion section. This will help remind the reader of these issues. This will also facilitate comparisons with other literature results (for example, the detailed data in Gidden et al. 2019 for each scenario.).

Line 133 - "the similar characteristics" -> "similar characteristics"

Line 141 - The assumptions behind the RCPs were not "homogeneous" (each RCP was produced by different models, and assumptions were not harmonized between models). The assumptions would be more accurately described as "relatively similar", or some such wording.

Line 156 "increased global ammonia (NH3) emissions (not shown)," (would be useful to reference Gidden et al. 2019 Figure E3 here as these are shown there.)

Line 187 "However, towards the end of the century North Africa and the Middle East reaches similar levels." Its not clear what this line means, since there is no one behavior for this region. For SO2 (Figure 2), NAF-MDE either stays at current levels, or declines (depending on scenario), while BC either increases somewhat, or declines.

Line 273: "whereas the mainly residential, and therefore more challenging, BC sources remain largely unchecked, the aerosol forcing may follow a different path than estimated here"

This is too oversimplified, since residential sources also emit copious amounts of OC, which means that the net forcing from residential sources depends on the balance between BC/OC emissions, and the relative per Tg forcing of each in any particular model. The result is that the net forcing from residential emissions is quite uncertain, likely even as to sign (particularly since rapid adjustments reduce the impact of BC), and trends even more so.

Line 307 " find no evidence that aerosol emissions reductions drive a particularly rapid near-term warming in this scenario. " Perhaps point out here that this points to the significant inter-model differences in aerosol response.

Line 331 "impose" is an odd word here, perhaps "drive"?

general: It would make this work more helpful for readers if the time series of global and regional forcing could be provided in the supplement.

Also, forcing by species (sulfate, nitrate, BC, OC, etc.) (+ aerosol cloud interactions)

should also be provided. These are, in part, discussed in the manuscript, but a table with numerical values should be provided.

---

## Author Comment (AC1) · 23 Sep 2019

Response to review of *"Anthropogenic aerosol forcing under the Shared Socioeconomic Pathways"* by Marianne T. Lund, Gunnar Myhre, and Bjørn H. Samset.

We thank the anonymous referee #1 for the careful and thorough review of our paper, and the useful suggestions. Responses to individual comments are given below.

L18-19: The differences between 2100 and 2015 reported here should equal the 1750-2100 values (given in L14-15) minus the 1750-2015 value (-0.61 W/m2) I'd expect, but they don't. Seems like a mistake and accidentally the opposite of the 1750-2100 values are given here instead of the differences for 2100 vs 2015 which I'd think are +0.57 and +0.10 W/m2, respectively (unless the values in L14-15 are wrong). Or if there is some non-linearity or incompatibility between the various estimates please revise so this is not so confusing to the reader.

We thank the reviewer for noticing this error. The mistake here is the value reported for 1750-2015, which was misplaced by the 1750-2014 number. The correct value is -0.55 Wm-2. Furthermore, the reference to 1750-2014 numbers in the results sections has been changed to 1750-2015 for consistency.

L43-44: Rather than the Li et al study cited here for air quality/health, which looked only at one country using one model, one of the worldwide multi-model studies by Silva et al would be a better choice (e.g. Silva et al, The effect of future ambient air pollution on human premature mortality to 2100 using output from the ACCMIP model ensemble, Atmos. Chem. Phys., 16, 9847-9862, 2016.)

Similarly, rather than (or in addition if you like) the Szopa et al and Westervelt et al studies, each of which looked at just one model, the reader could more usefully be pointed to the multi-model studies such as Shindell et al., Radiative forcing in the ACCMIP historical and future climate simulations, Atmos. Chem. Phys., 13, 2939–2974, 2013; and Rotstayn et al, Why does aerosol forcing control historical global-mean surface temperature change in CMIP5 models?, J. Climate, 28, 6608-6625, 2015.

Good suggestions, both citations have been added.

L56: Clearer to write as "continue to be high and are increasing" than "also continue to be high and increasing" as the 'also' seems not to relate to the previous sentence (about SO2) but an earlier one so is hard to follow.

Sentence modified.

L56-58: It would be good to include additional explanation here that the RCP aerosol emissions were based on the assumption that economic growth leads to decreasing emissions of precursors, a so-called environmental Kuznets' curve/behavior. An excellent discussion of the background to assumptions such as those is given in Amann et al, 2013. Regional and global emissions of air pollutants: recent trends and future scenarios. Annual Review of Environment and Resources, 38, pp.31-55; and a recent paper analysed the link between economic growth and aerosol precursor emissions and indeed supports the discussion here (and in Amann et al) that RCP projections are likely too optimistic as those emissions do not necessarily decline with growth in GDP or follow CO2 (see Ru et al, The long-term relationship between emissions and economic growth for SO2, CO2 and BC, Env. Res. Lett., 13, 124021, 2018.) This discussion would help set the context for the new approach in the SSPs where air pollution controls have independent settings, as described in the next paragraph.

An important issue that we did not originally think to mention. The text has been expanded with an additional paragraph and now reads:

*"The aerosol and precursor emissions in the RCPs are generated following the assumption that economic growth leads to decreased emissions using the so-called environmental Kuznets curve. This real-world representativeness of this relationship has, however, been questioned (Amann et al., 2013; Ru et al., 2018). This, combined with the slow observed progress on alleviating air pollution, raises the question of whether previous projections of future emissions are too optimistic in terms of pollution control."*

L74: Replace 'there is progress is slowed' with just 'progress is slowed'.

Corrected.

L90: Please also give the number of vertical layers and the model top as that'd help the reader get a general sense of the model in addition to the horizontal resolution.

Text modified to

*"Here the model is run in a 2.25°x2.25° horizontal resolution, with 60 vertical levels (the uppermost centered at 0.1 hPa)."*

L150: 'Becomes' -> 'become'.

Corrected.

L186: Delete 'the' before 'South Asia'.

Corrected.

L189: Add 'the' before 'Sahara'.

Corrected.

L201: Add 'that' after 'demonstrated'.

Corrected.

L220-221: The authors describe the conclusion drawn in the Stjern et al study about the semi-direct impact of BC here. They should also include the results of Allen et al., Observationally constrained aerosol–cloud semi-direct effects, npjCAS, 2, 16, 2019 as those suggest the semi-direct effect may not be well captured by the models used in Stjern et al and so drawing conclusions about the sign of the adjustment is less certain than the current text implies.

We have modified the text to account for both these studies and their contrasting results:

*"Using data from several global models, Stjern et al. (2017) found that the rapid adjustments by clouds offset a significant fraction of the aerosols' positive RFari, reducing the net BC climate impact. A recent study by (Allen et al., 2019) instead found a positive cloud rapid adjustment The latter finding would imply a much stronger non-cloud negative rapid adjustment than presented in (Smith et al., 2018) and methodological differences clearly need to be better resolved in order to understand the contrasting results."*

The same goes for text in L281-282 – might reduce temperature response not the more definitive 'have been shown to'.

Modified.

L225-230: It would be helpful to be clear about the time periods for the forcing values quoted from these other two studies (for Fiedler, forcing when vs when; for Partanen, it is stated that it's for 2100, but relative to when?).

Both studies provide end-of-the-century values (mid-2090s and 2100, respectively) relative to 1850. The text has been modified accordingly.

L269-282: The discussion here starts to delve into the role of individual aerosol species, which is good. I like the current discussion regarding SO2 and BC. I'd like to see a bit more on this, however, in particular I would request that the authors extend their Table S1 to include the year 2100 (data they should already have), and add a discussion of the nitrate forcing towards the end of the century given the large increase in burden shown in Figure 1h for SSP3. That could be usefully compared with the results of the Bauer et al, Bellouin et al, and Shindell et al (see reference in comments on L43-44 for the latter, the other two are already cited) studies that discussed the possibility of substantial increases in negative RF from nitrate over the 21st century.

We have expanded the discussion, including a paragraph about nitrate and, guided by the comments by referee #2, uncertainties related to BC and co-emitted OA.

L318-320: Again could compare with the Silva et al results for future RCPs (see reference in comments on L43-44).

Text added:

*"Silva et al. (2016) found avoided premature mortality in 2100 of between -2.39 and -1.31 million deaths per year for the four RCP."*

L331: 'Impose' seems an odd word to use here, as usually things people don't want are imposed upon them whereas improved air quality is something people do want. I suggest changing to 'lead to' or something similar.

Good point. Text modified.

L332-347: The Methods section describes how only cloud albedo effects are analysed here (as RFaci is calculated offline). This section should point out that the calculations here not only neglect climate feedbacks, but neglect the entire cloud lifetime portion of RFaci, which may be important.

Added:

*"Our estimates of RFaci exclude contributions from cloud lifetime changes. The estimates of cloud lifetime effect are generally lower in recent studies than in early work, but still give non-negligible contribution to the aerosol forcing (Storelvmo, 2017)."*

L363: Useful for those not so familiar with the SSPs to add "(SSP3)" after "regionally fragmented world with slower mitigation progress" here.

Agree, added.

L365: "Increases" -> "increase" and "starts" -> "start".

Corrected.

---

## Author Comment (AC2) · 23 Sep 2019

Response to review of *"Anthropogenic aerosol forcing under the Shared Socioeconomic Pathways"* by Marianne T. Lund, Gunnar Myhre, and Bjørn H. Samset.

We thank the anonymous referee #2 for the careful and thorough review of our paper, and the useful suggestions. Responses to individual comments are given below.

Abstract this text "aerosols under three different levels of air pollution control: strong (SSP1), medium (SSP2) and weak (SSP3). " should be revised, given that far more than air pollution controls impact emission levels in these three scenarios. It would be more accurate to describe these as representing three contrasting projections for air pollutant emission levels.

A valid point, we have modified according the referee's suggestion.

More context should be given in the introduction when the scenarios are introduced. It's too simplistic to simply call the scenarios simply high/low air pollution. Air pollution controls plus the magnitude of the various drivers of emissions (e.g., population levels, economic growth, rural access to modern energy, GHG emissions policy, etc.) all play a role in determining the ultimate emissions level. For example, referring to Rao et al. Figure 2, for the Ref case scenarios (e.g. no GHG emissions reduction policy) emissions can differ significantly between SSP1 and SSP5, even though both of these scenarios represent storylines with strong air pollution controls. Similarly, emission levels are generally quite a bit higher in SSP3 as compared to SSP4, even though the emission control assumptions are similar. Note also that the SSPs are from different projection models, which means that one also has be cautious in such comparisons.

We thank the reviewer for these reflections. For detailed descriptions of the assumptions underlying the scenarios and how they drive the differences in emissions, we refer to the cited literature. However, we have rewritten the paragraphs describing the projections to clarify the connection between the air pollution storylines and SSP baseline marker and climate mitigation scenarios, and to emphasize the complexity of the interplaying factors. While the strong/medium/weak terminology is kept for consistency with Rao et al, recognizing that this is a generalization, we include the high/medium/low challenges to mitigation and adaptation that characterizes the given SSP later in the text as well.

The impact of inter-annual variability should be discussed given that meteorology for just one year is used. How much does the selection of that year influence results?

Good point. The effect of different meteorological data sets on aerosol abundances in the OsloCTM3 was investigated in a recent documentation paper by Lund et al. (2018, GMD). Here we add a brief summary and add the following text:

*"All simulations are performed with meteorological data for 2010. Lund et al. (2018) investigated the impact of meteorology on the simulated aerosol abundances using data for two years with opposite El Niño–Southern Oscillation (ENSO) index. Differences in global burden of up to 10% for some aerosol species where found, with larger values in localized regions over the tropical Pacific and Atlantic Oceans."*

Line 47 - "generally reflect the assumption that stringent air quality regulations will be successfully implemented globally (Rao et al., 2017" Suggest replacing stringent with a somewhat more neutral

word (perhaps "substantial""). The RCP's represented a somewhat middle of the road air pollutant emission control assumptions, but by no means were they at maximally feasible levels (which is what might be read by "stringent"). The more important point to be made here is that there was limited variation in air pollution control assumptions across the RCP scenarios.

Modified to substantial. Based on a comment from referee #1 we have also added a paragraph about the underlying environmental Kuznets curve assumption, which limits the variation across RCPs, hence more explicitly addressing this point.

Line 95-105 What was assumed for open burning? These are also supplied in the future scenarios, (but are from van Marle et al 2017, not Hoesly et al., 2018). Note that these are not "natural" emissions, as much of these emissions are due to human activity.

We use the biomass burning emissions supplied by in the future scenarios. The "vegetation" emissions referred to as "natural" is biogenic VOCs. This has been specified in the text now.

Line 123 While I understand why this "For simplicity we refer to SSP1-1.9 as SSP1, SSP2-4.5 as SSP2 and SSP3-7.0 as SSP3 throughout the text." is done, however, this is inaccurate and may lead to misunderstanding on the part of readers, as there can be systematic differences even between scenarios with the same storyline. SSP1-1.9, for example, is a very strong GHG mitigation scenario which means that fossil fuel use is drastically reduced (and what fossil fuel that is used tends to have lower air pollutant emissions). So emissions will tend to be on the low side of what is already a low reference scenario. Emissions can be much lower than the reference case SSP1, particularly in earlier years. Similarly for SSP2-4.5, emissions here can be significantly influenced by the fact that the 4.5 scenario contains policies to limit greenhouse gas emissions. I suggest first, as mentioned above, that a little additional context be given for these scenarios. It would help, on first introducing the scenarios, that the fuller version of the scenario name that also contains the model name is used. That will help enforce to the readers that these are from different projection models. The presence, or not, of a climate policy should be mentioned (e.g. present in SSP1-1.9 and SSP2-4.5) should be mentioned in this introduction, since this can have a strong influence on the emissions pathway. Some of the figures have the fuller scenario names and some do not. Suggest that all figures have the names that contain the forcing target. I suggest the fuller scenario names (e.g., SSP1-1.9) be returned to in the discussion and conclusion section. This will help remind the reader of these issues. This will also facilitate comparisons with other literature results (for example, the detailed data in Gidden et al. 2019 for each scenario.).

We see the point and have included full names with forcing target throughout the text. We have also expanded the introduction and methodology sections.

Line 133 - "the similar characteristics" -> "similar characteristics"

Corrected.

Line 141 - The assumptions behind the RCPs were not "homogeneous" (each RCP was produced by different models, and assumptions were not harmonized between models). The assumptions would be more accurately described as "relatively similar", or some such wording.

Modified.

Line 156 "increased global ammonia (NH3) emissions (not shown)," (would be useful to reference Gidden et al. 2019 Figure E3 here as these are shown there.)

Included.

Line 187 "However, towards the end of the century North Africa and the Middle East reaches similar levels." Its not clear what this line means, since there is no one behavior for this region. For SO2 (Figure 2), NAF-MDE either stays at current levels, or declines (depending on scenario), while BC either increases somewhat, or declines.

Similar to South and East Asia. Text has been clarified and now reads:

"Towards the end of the century North Africa and the Middle East are projected to experience levels similar to those in South and East Asia."

Line273: "whereas the mainly residential, and therefore more challenging, BC sources remain largely unchecked, the aerosol forcing may follow a different path than estimated here" This is too oversimplified, since residential sources also emit copious amounts of OC, which means that the net forcing from residential sources depends on the balance between BC/OC emissions, and the relative per Tg forcing of each in any particular model. The result is that the net forcing from residential emissions is quite uncertain, likely even as to sign (particularly since rapid adjustments reduce the impact of BC), and trends even more so.

As noted, this is a highly illustrative case meant to initiate discussion around whether there's a possibility that emissions of different species may follow different pathways in the coming decades. The RF estimates provided also include both OA and nitrate, and the rapid adjustments of BC are noted. However, following this comment and input from the anonymous referee #1, we have expanded the discussion with more on the role of nitrate, as well as the uncertainties surrounding the forcing of OA, which could also have a non-negligible absorption (brown carbon).

Line 307 " find no evidence that aerosol emissions reductions drive a particularly rapid near-term warming in this scenario. " Perhaps point out here that this points to the significant inter-model differences in aerosol response.

Yes, good suggestion. Included.

Line 331 "impose" is an odd word here, perhaps "drive"?

Modified to "lead to"

general: It would make this work more helpful for readers if the time series of global and regional forcing could be provided in the supplement. Also, forcing by species (sulfate, nitrate, BC, OC, etc.) (+ aerosol cloud interactions) should also be provided. These are, in part, discussed in the manuscript, but a table with numerical values should be provided.

Full time series of total RFari and RFaci has been added. Forcing by species is only available for the direct aerosol effect and for the selected years presented in Table S1.